# Beauty of symmetry - The impact of logo symmetry on perceived product quality

**Xianghu Wu**[ID]*

Southwest Jiaotong University Hope College, Chengdu, China

* Xianghu_Wu_0802@163.com

## Abstract

In an era of intense brand competition, a successful logo can effectively boost consumer awareness of a company. However, existing research has not thoroughly examined the aspect of symmetry in logo design. Addressing this gap, the present study investigates the impact of logo symmetry on consumers' perceived product quality. Through three online experiments, the results revealed that symmetrical logos significantly enhance perceived product quality compared to asymmetrical logos, with perceived stability mediating this effect. Furthermore, the study confirmed the moderating role of cognitive load, showing that symmetrical logos positively affect perceived product quality only under low cognitive load, while this effect vanishes under high cognitive load. This study contributes to the field of visual logo design and provides practical insights for companies in product promotion and logo design.

**Citation:** Wu X (2025) Beauty of symmetry - The impact of logo symmetry on perceived product quality. PLoS ONE 20(1): e0317229. https://doi.org/10.1371/journal.pone.0317229

**Data Availability Statement:** All relevant data are within the paper and its Supporting Information files.

**Funding:** The author(s) received no specific funding for this work.

## Introduction

In today's highly competitive market, brand logos have become crucial assets for enterprises [1] A well-designed, eye-catching logo not only attracts consumers' attention but also leaves a lasting impression, thereby enhancing brand recall [2]. When consumers first encounter a brand, they often form an initial impression within seconds, with the logo playing a pivotal role in shaping this perception [3]. More than just a visual identifier, a brand logo embodies a company's culture, values, and market positioning, significantly impacting the enterprise's overall success [4]. Logos have been found to influence brand recognition [5] and greatly affect consumer trust and loyalty [6]. In the era of digital marketing, where visual impact and information quality are paramount, the design elements of logos have become even more significant.

One of the key factors in logo design is symmetry. Symmetrical designs are often perceived as more attractive, balanced, and credible [7]. This preference for symmetry is rooted in cognitive processing; symmetrical visuals are easier to process, leading to a perception of harmony and stability [8]. Therefore, exploring how logo symmetry influences consumers' perception of product quality is of both theoretical and practical importance, as it can help businesses create logos that foster positive consumer evaluations and brand loyalty [9]. Current research on logo design emphasizes its role as a common visual element in brand communication, conveying

**Competing interests:** The authors have declared that no competing interests exist.

corporate values and the brand image [4]. Logo design includes various elements such as shape, color, size, position, and style [10–13]. Scholars have extensively studied these elements and their influence on consumer perception. However, few studies have investigated how logo symmetry affects consumers' perceptions of enterprises, particularly regarding product quality. Existing research on logo symmetry primarily focuses on its aesthetics and symbolism [9], its positive impact on consumer expectations [14, 15], and the values it conveys [4]. The impact of logo design on tangible product attributes, such as perceived quality, remains underexplored.

This study seeks to address this gap by examining how logo symmetry influences consumers' perception of product quality. Specifically, this study aims to reveal the mechanisms through which symmetrical logos impact perceived product quality. By conducting three experiments, this study investigate whether symmetrical logos enhance product quality perceptions compared to asymmetrical logos, with perceived stability acting as a mediating factor. Additionally, this study explore whether cognitive load moderates this effect - specifically, whether logo symmetry influences perceived quality only when consumers experience low cognitive load.

Our findings contribute to the literature on logo design and visual perception in several ways. First, this study demonstrate that logo symmetry directly impacts consumer perception of product attributes, providing insights into how visual elements influence consumer evaluations beyond mere aesthetics. Second, this study introduce perceived stability as a mediating factor, expanding the understanding of how logo design affects consumer behavior. Finally, this study highlight the role of cognitive load in moderating the effect of logo symmetry, suggesting that the effectiveness of visual design elements can vary depending on the cognitive context of consumers.

From a practical perspective, this study provides actionable insights for marketers and designers. Companies can enhance consumers' perceived quality of their products by adopting symmetrical logos, especially for products that need to convey stability and trust [9]. Furthermore, marketing strategies should consider the cognitive load of the target audience - symmetrical logos are more effective when consumers are not overwhelmed by competing stimuli. This study also opens avenues for future studies to explore the impact of other visual elements, such as color and shape complexity, on consumer behavior, particularly across different cultural contexts. In summary, by investigating the impact of logo symmetry on perceived product quality and the mechanisms behind this effect, this study aims to advance the understanding of logo design's role in consumer perception. These findings are not only theoretically significant but also offer practical guidance for effective marketing and branding strategies in today's visually driven marketplace.

## Literature review

### Symmetry in design

Symmetry is a fundamental concept that plays a crucial role in various fields such as psychology, marketing, and mathematics. Defined by the relative arrangement of components within a composition, symmetry represents an essential holistic property [16]. It is prevalent in nature, with examples like bilateral symmetry in animals and flowers, which are not only common but also aesthetically pleasing [17] Studies have shown that symmetry influences consumer responses, driven by processing fluency and aesthetic appeal [18]. In facial characteristics, symmetrical structures can dilute judgments, but the addition of authentic cues like freckles can enhance perceptions and attitudes [19]. Overall, symmetry is a multifaceted concept impacting perception, aesthetics, and consumer behavior across various domains.

Previous research has demonstrated that strategic use of symmetry can positively influence brand perception and consumer engagement. Symmetry impacts consumer responses through processing fluency and aesthetic appeal [16]. It is recognized as a key visual design principle affecting consumer preferences alongside complexity [20]. In graphic design, incorporating symmetrical elements and higher image contrast can increase consumer engagement and liking, especially in social media contexts [18]. By leveraging the aesthetic appeal and cognitive fluency associated with symmetry, designers can develop visually compelling and effective communication materials.

Researches on logo symmetry had shown a positive relationship between visual harmony, including symmetry, and subjective ratings of logos, leading to better recognition [16]. Additionally, brand logo symmetry influences consumer inferences and perceptions of product design [9]. Interestingly, there is a connection between political conservatism and preferences for (a)symmetric logos, suggesting ideological influences on design choices [21]. These findings highlight the significance of logo symmetry in shaping brand personality, consumer perceptions, and preferences.

Despite extensive research on symmetry's impact on consumer responses, a notable limitation is the lack of focus on how logo symmetry specifically affects consumer-perceived product quality. While studies have highlighted symmetry's role in enhancing visual harmony, recognition, and brand perception, the direct influence of logo symmetry on product quality assessments remains underexplored. This gap in the literature indicates a need for targeted research to understand how logo symmetry influences perceived product quality, providing valuable insights for marketers and designers.

## Perceived stability

Consumer perceived stability refers to the extent to which consumers feel consistency, stability, and dependability in their interactions with a brand or service. This perception can be influenced by brand loyalty, risk reduction, and the establishment of trust over time [22]. Studies have shown that perceived stability plays a crucial role in shaping consumer behavior, affecting purchase intentions and relationship quality [23]. Beyond individual transactions, the concept of stability also encompasses broader notions of social and moral stability, influencing how consumers engage with brands and products [24]. Overall, consumer perceived stability is a multifaceted construct that includes elements of trust, consistency, and stability in consumer-brand interactions.

Brand stability signals a brand's attributes as permanent, allowing customers to anticipate its future performance [22]. Salesperson perceived stability impacts sales force performance, highlighting the importance of stability in customer interactions [23]. Responses to negative customer reviews can alter perceived stability, affecting trust and purchase intentions [25]. Stability is also linked to lasting product performance, where round numbers activate associations of stability, influencing perceptions of product benefits [26]. These studies collectively emphasize the significance of stability in consumer psychology and marketing strategies.

## Hypothesis development

### Logo symmetry and perceived quality

Symmetrical logos are fundamental in enhancing visual stability, making them inherently more appealing and leading to a more positive perception among consumers. Empirical studies have consistently demonstrated that symmetrical designs are perceived as more aesthetically pleasing, thereby significantly influencing consumer interactions with logos and brand elements [16]. The concept of symmetry extends beyond mere visual attractiveness; it conveys

attributes such as balance, order, and stability, which collectively contribute to a positive perception of a brand's intrinsic values. This preference for symmetry indicates that symmetrical logos evoke a sense of coherence, elegance, and high quality, which ultimately leads to more favorable consumer attitudes and a deeper emotional connection with the brand [20]. For example, the logos of brand like Toyota is quintessential example of symmetry that convey these positive associations, contributing to their strong brand identities. Furthermore, the consistent use of symmetrical logos fosters brand identity, reinforcing consumer trust and loyalty by evoking a sense of familiarity and stability [27]. In addition, symmetry is frequently associated with permanence and reliability - key attributes that are crucial for brands seeking to establish long-term relationships with consumers, such as in the case of the symmetrical logo of McDonald's, which conveys reliability and endurance [26].

A substantial body of literature has demonstrated that stable visual elements significantly enhance consumer perceptions of product quality, durability, and shelf life. For instance, Coelho et al. (2019) illustrate how stable shapes contribute not only to a positive brand image but also to an enhanced perception of product quality [27], while Pena - Marin and Bhargave (2019) argue that products perceived as having stable attributes are evaluated more favorably by consumers [26]. This can be observed in the packaging of high-end perfumes, which often use sturdy, symmetrical bottles to convey a sense of premium quality and stability. Similarly, Yan et al. (2013) noted that smaller, meticulously designed products are often seen as more challenging to manufacture, thereby implying higher quality [28]. Applying this concept to stable shapes, products that visually signal stability—such as the iconic Coca-Cola glass bottle— are often perceived as having higher quality. Mitra & Golder (2006) highlight the impact of objective quality on consumer perceptions [29], emphasizing that stability in design fosters consumer trust and loyalty [22]. For example, the durable and stable design of Tupperware products conveys longevity and reliability, which positively influences consumer perceptions of quality and durability. In conclusion, products with stable visual elements are more likely to enhance consumer perceptions of quality, significantly shaping evaluations of product longevity and fostering a positive brand image.

Thus, the following hypotheses are proposed:

H1: Compared with asymmetrical logo, symmetrical logo symmetry increases consumer perceived quality.

H2: Perceived stability mediates the role of logo symmetry on consumer perceived quality.

## Moderator role of cognitive load

Cognitive load refers to the mental effort required to process information in working memory. When cognitive load is low, individuals have greater cognitive resources available to engage with and appreciate complex stimuli, such as the symmetry of logos [30]. Conversely, under conditions of high cognitive load, these cognitive resources are largely occupied, which can hinder the ability to process intricate visual details. For example, consumers shopping online during a relaxed browsing session (low cognitive load) may be more capable of appreciating a brand logo's symmetrical design than when multitasking or in a rushed environment (high cognitive load). Research suggests that high logo symmetry is associated with higher perceived quality under low cognitive load, whereas this effect diminishes under high cognitive load. This is supported by findings that cognitive load influences visual perception and attention mechanisms. Lavie et al. (2004) and Konstantinou and Lavie (2013) demonstrated that individuals have greater capacity to process visual details, such as symmetry, under low cognitive load conditions, leading to enhanced perception and evaluation of these stimuli [31, 32].

Moreover, empirical studies by Saiphoo and Want (2018) and Derpsch et al. (2021) indicate that cognitive load plays a significant role in shaping the evaluation of visual information, further supporting the notion that symmetry is more effectively appreciated when cognitive resources are not heavily taxed [33, 34]. In a practical context, this suggests that brands aiming to convey a sense of quality through logo design should consider the cognitive state of their target audience. For example, advertising campaigns during times when consumers are likely to be relaxed (e.g., during weekends or leisure activities) may enhance the positive effect of symmetrical logos on perceived quality.

Thus, the following hypotheses are proposed:

H3: cognitive load moderates the effect of logo symmetry on perceived quality. Specifically, When consumers were in low cognitive load condition, compared with asymmetrical logo, symmetrical logo symmetry increases consumer perceived quality. However, consumers were in high cognitive load condition, this effect was dismissed (Fig 1).

## Methodology

### Overview of study

We conducted three studies to validate the proposed hypotheses through online experiments. Study 1 preliminarily validated Hypothesis 1 using consumers from Credamo, indicating that logo symmetry affects consumers' perceived product quality. Study 2 examined the underlying mechanism of this effect, showing that a symmetrical logo enhances consumers' perceived stability, which further increases their perceived quality. Study 3 explored the boundary conditions of this effect, revealing that when consumers are in a state of cognitive load, the influence of logo symmetry on perceived quality disappears.

### Study 1

**Purpose.**   Study 1 aimed to test H1, which is compared with low logo symmetry, high logo symmetry increases consumer perceived quality. This study conducted an online experiment.

**Stimulus and pretest.**   Referring to the study by Bettels and Wiedmann (2019), Study 1 selected a set of symmetrical and asymmetrical logos as experimental materials [9]. In the

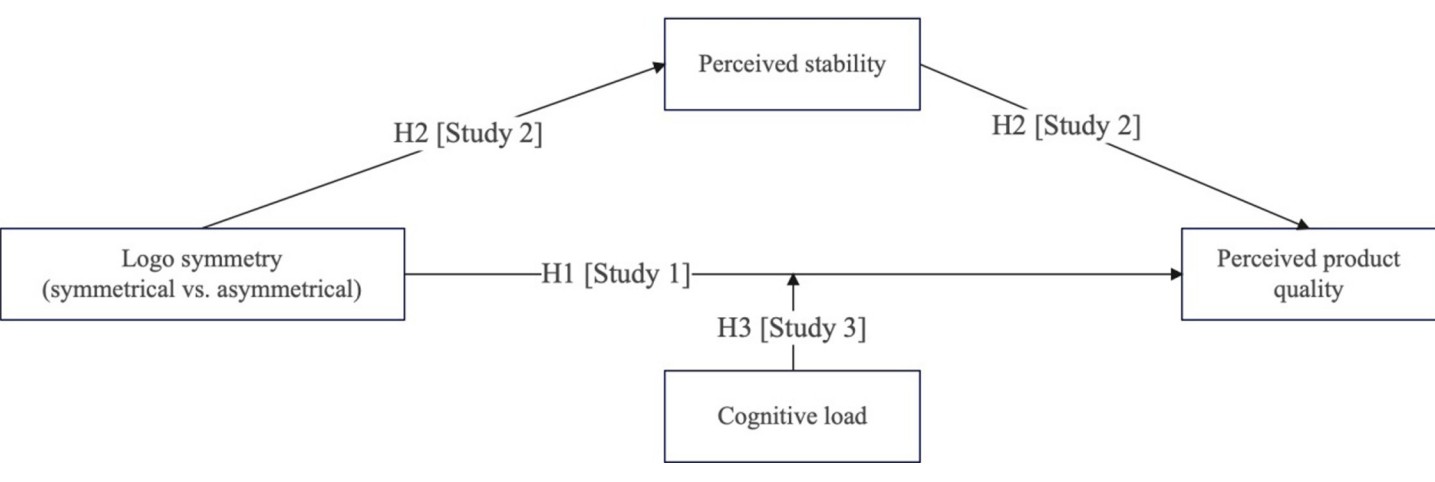

**Fig 1. Research framework.**

symmetry condition, participants viewed a set of line designs symmetrically about a vertical centerline; whereas in the low symmetry condition, participants viewed a set of logos where one half matched the symmetrical condition, but the other half consisted of randomly designed lines (Fig 2).

An independent pretest conducted on the Credamo platform (N = 66, 53.33% female, $M_{age}$ = 37.25) indicated significant differences in perceived symmetry between the two sets of logos. The perceived symmetry of the symmetrical logos (M = 5.15, SD = 1.70) was significantly higher than that of the asymmetrical logos (M = 2.06, SD = 1.22; $F (1, 64)$ = 72.000, $p < .001$). However, there were no significant differences in participants' liking of the logos ($F (1, 64)$ = .006, $p = .937$) or aesthetic perception ($F (1, 64)$ = 1.210, $p = .275$). These results suggest that the experimental materials are suitable for use in the main study.

**Participants and procedure.** 132 participants (54.55% female, $M_{age}$ = 38.38) completed this study via Credamo (a Chinese data collection platform; www.credamo.com) for a nominal payment and they were randomly assigned to either asymmetrical logo or symmetrical logo conditions. Following previous studies [9], participants will view an advertisement for a chair brand, described as "Our brand has long been committed to producing seats that satisfy consumers, while we also pay great attention to the quality of our products." The brand logo and the chair will appear on the same page, with the product positioned slightly to the left of the logo. After viewing the advertisement, participants will be asked to rate the perceived quality of the product using a 4-item, 7-point scale adopted from Cinelli et al. (2020) (e.g. "How high in quality do you think this brand is? from 1 = 'very low quality' to 7 = 'very high quality'", Alpha = .918). Additionally, participants will report their familiarity with the brand logo ("How familiar are you with this brand's logo?" from 1 = 'Not familiar at all' to 7 = 'Very

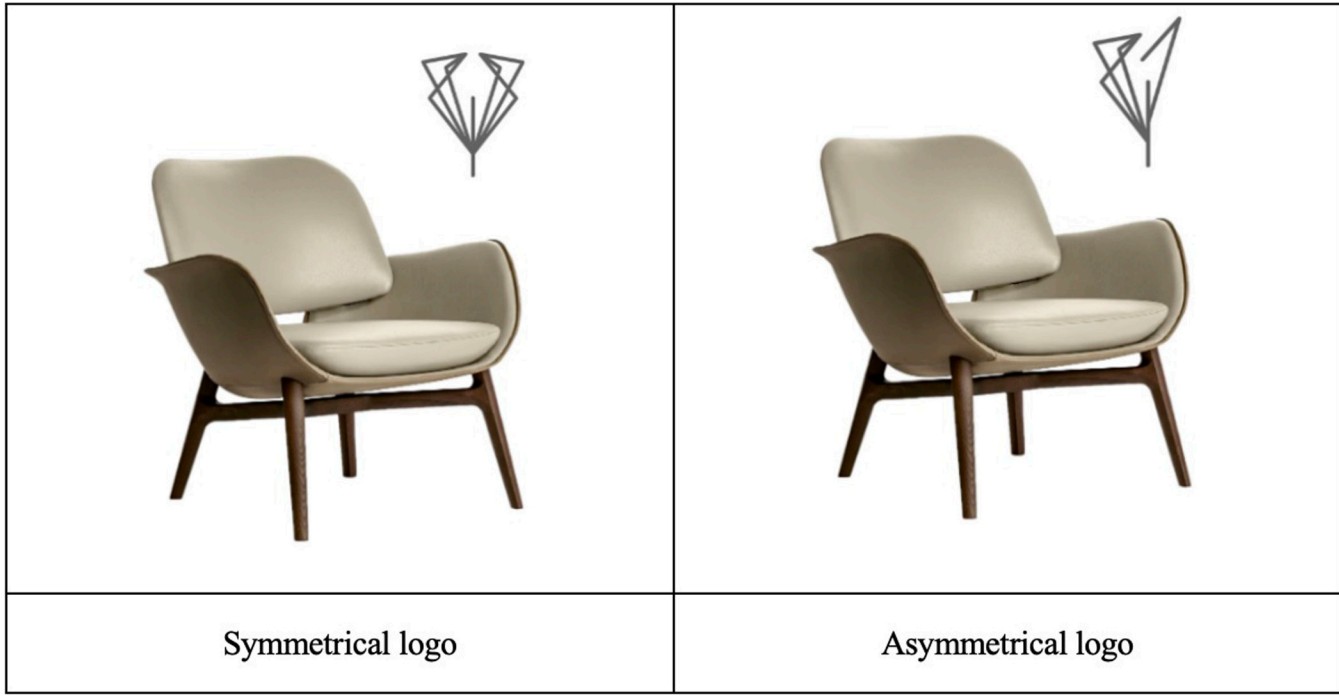

**Fig 2. Stimuli using in Study 1.**

familiar'), their liking of the product itself ("How much do you like this product?") and manipulation check ("How symmetrical do you find this logo?" from 1 = very asymmetrical to 7 = very symmetrical). Demographic information was collected from participants upon concluding the main experiment.

**Result.** *Perceived product quality*. A one-way ANOVA indicated that participants in the symmetrical condition perceived higher product quality (M = 4.92, SD = 1.69) than those in the asymmetrical logo condition (M = 4.19, SD = 1.33; F (1, 130) = 7.682, $p$ = 0.006), which is consistent with the prediction. Moreover, with logo familiarity and product liking as covariates, the results of a one-way ANOVA indicated that the effect of a symmetrical (vs. asymmetrical) logo on perceived product quality remains significant (F (1, 128) = 38.330, $p$ < .001; Fig 3).

**Discussion.** Study 1 employed an online experiment to preliminarily verify the effect of logo symmetry on perceived quality. The results indicated that compared to asymmetrical logos, symmetrical logos enhance consumers' perceived product quality. While Study 1 directly verified the main effect of logo symmetry on perceived product quality, it did not investigate the underlying mechanisms of this effect, leaving the explanatory scope incomplete. Therefore, Study 2 aimed to further explore the potential mediating processes. This study hypothesizes that the positive influence of logo symmetry on perceived product quality is mediated by perceived stability; specifically, a symmetrical logo may lead consumers to form a higher perception of stability, which in turn enhances perceived product quality.

## Study 2

**Purpose.** Study 2 tested the underlying mechanism of logo symmetry on perceived quality, specifically focusing on perceived stability. Additionally, Study 2 ruled out the alternative explanation of perceived naturalness, as symmetrical designs might exhibit higher naturalness [35], which could lead consumers to perceive higher usable quality. Specifically, this study used the same stimuli as in Study 1, further extending the applicability of Study 1.

**Participants and procedure.** 168 participants (59.52%%, $M_{age}$ = 37.54) were recruited from Credamo and completed this study. They were randomly assigned to one of the two conditions of a single-factor (logo symmetry: symmetrical logo vs. asymmetrical logo) between-subject design. Participants were informed: "Currently, a wooden chair manufacturing and

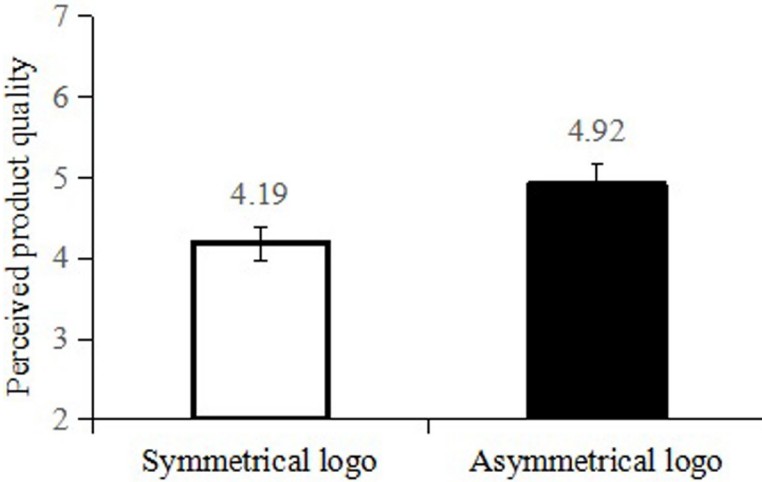

**Fig 3. The result of Study 1.**

sales company is planning to revise its logo and seeks consumer evaluations for the new design."

After completing the consent form, participants were presented with an advertisement for a mobile suitcase featuring the same logo as in Study 1 (Fig 2). After viewing the advertisement, participants responded to three items on a 7-point scale measuring perceived stability ("I think this product is stable," "I think this product is durable," and "I think this product is balanced" [26]), as well as one item on a 7-point scale measuring perceived naturalness ("How natural do you think this product is?" from 1 = "more artificial" to 7 = "more natural than the other product" [36]). They were also asked to indicate their perceived quality of this product, as in Study 1, along with their familiarity, liking, and manipulation check. Demographic information was collected from participants upon concluding the main experiment.

**Result.** *Manipulation check.* An ANOVA on manipulation check indicated that participants in the symmetrical logo condition perceived higher symmetry (M = 5.27, SD = 1.90) than those in the asymmetrical logo condition (M = 2.30, SD = 1.27; $F(1,166) = 142.209$, $p < .001$), suggesting a successful manipulation.

*Perceived quality.* An ANOVA on participants' perceived quality indicated that participants in the symmetrical logo condition perceived higher quality (M = 4.63, SD = 1.75) than those in the asymmetrical logo condition (M = 3.71, SD = 1.56; $F(1, 166) = 12.834$, $p < .001$), Additionally, an ANOVA analysis was conducted using participants' familiarity with and liking of the logo as covariates. The results indicated that the effect of logo symmetry on perceived quality remained significant ($F(1,164) = 57.992$, $p < .001$), further supporting Hypothesis 1.

*Perceived stability and perceived naturalness.* ANOVAs on perceived stability and perceived naturalness indicated that consumers in the symmetrical logo group reported higher perceived stability ($F(1, 166) = 5.888$, $p = .016$) compared to those in the asymmetrical logo group, which aligns with the expected hypothesis. Additionally, there was no significant effect on perceived naturalness ($F(1, 166) = 0.176$, $p = .675$), which may be due to the naturalness of the symmetrical logo not transferring to product attributes. Furthermore, this study conducted a mediation effect test on perceived stability using Hayes PROCESS.

*Mediation analysis.* To test the mediating role of perceived stability, and further rule out the alternative explanations of perceived naturalness, mediation analyses with these variables as mediators were further conducted by adopting the bootstrapping procedure (5,000 samples, PROCESS model 4; [37]). The results showed a 95% confidence interval for perceived stability that excludes zero (95%CI = [.0744, .7561]; Fig 4), suggesting a significant mediation effect; however, for perceived naturalness, the 95% confidence intervals include zero (95%CI = [-.2011, .1186]) (see Fig 2). Further, including perceived naturalness as covariate in the

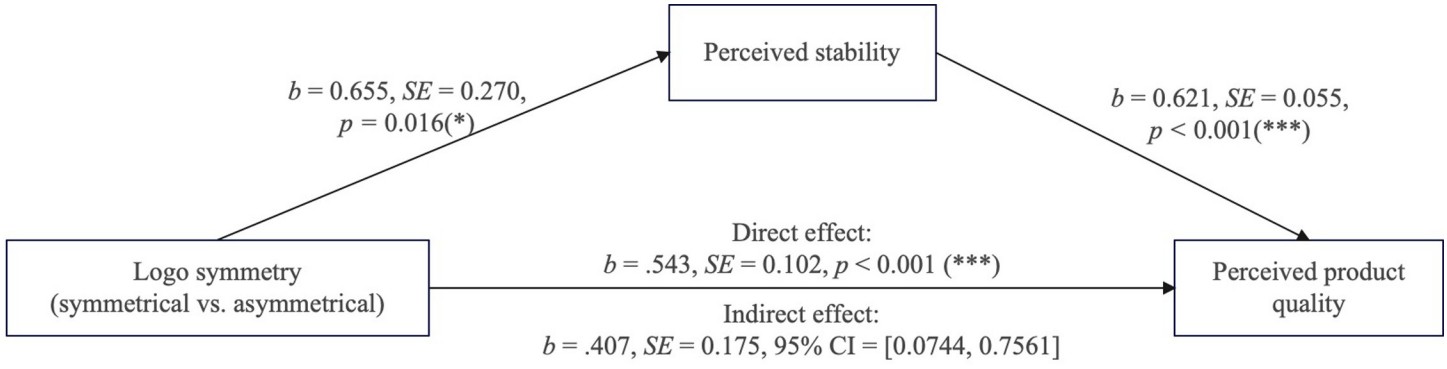

**Fig 4. The mediating role of perceived stability.**

mediation analysis also revealed a significant mediation effect of perceived stability (95% CI = [..2943, .6544], excluded zero). Therefore, these findings supported H2 and ruled out perceived naturalness for the effect of logo symmetry on perceived quality.

**Discussion.** Study 2 examined the mediation effect of perceived stability through an online experiment and further ruled out the alternative explanation of perceived naturalness. The results indicated that consumers perceived greater stability in products with symmetrical logos, thereby enhancing the perceived quality of the product. Drawing on the results from Studies 1 and 2, this research confirmed both the positive impact of logo symmetry on perceived quality and the underlying mechanism of this effect. In Study 3, this study investigated the moderating effect of this influence to strengthen the relevance of the findings within practical marketing contexts.

## Study 3

**Purpose.** The purpose of Study 3 was to verify the moderating effect of cognitive load. This study predicted that consumers would form a perception of quality through cognitive processes when faced with symmetrical logos. However, when consumers are under high cognitive load, this processing is hindered, potentially preventing the formation of an accurate perception of product quality. Thus, when consumers are under high cognitive load, the effect of logo symmetry on perceived product quality is expected to diminish.

**Stimulus and pretest.** For the experimental stimuli in Study 3, this study selected luggage, with the logo design based on the study by Bettels and Wiedmann (2019) [9]. This study created a symmetrical logo by arranging a group of triangles with alternating black and white colors (Fig 5). Jiang et al. (2016) found that acute elements can enhance perceived stability [10]. To avoid this influence, the asymmetrical logo design also incorporated the same acute elements. An independent pre-test conducted on Amazon Mechanical Turk (MTurk) (N = 61, 50.82% female, $M_{age}$ = 40.78) examined the manipulation check of logo symmetry. The results revealed that participants in the symmetrical logo condition perceived higher symmetry (M = 4.87, SD = 2.05) than those in the asymmetrical logo condition (M = 2.90, SD = 1.37; F(1, 59) = 19.393, p < .001).

**Participants and procedure.** This study conducted a 2 (logo symmetry: symmetrical logo vs. asymmetrical logo) × 2 (cognitive load: high vs. low) between-subjects design. Two hundred participants from MTurk ($M_{age}$ = 41.15, 67.50% female) took part in this study for monetary compensation and were randomly assigned to one of the four conditions. After completing the consent form, participants underwent the cognitive load manipulation process, based on the study by Jiang et al. (2016). Participants in the high cognitive load group were asked to memorize eight random digits "81097511", while those in the low cognitive load group were asked to memorize two digits "45".

Then participants were presented with an advertisement for a luggage product, introduced as "Thank you for participating in our luggage product survey. Below is our product logo. Please evaluate our product." To ensure continuity of the cognitive load process, this page required a minimum response time of 30 seconds. After viewing the advertisement, participants were asked to recall the memorized digits and complete the same perceived product quality measurement as in Study 1. Subsequently, they answered questions about their familiarity, liking, and a manipulation check of cognitive load ("How much effort did you put into completing the above task?" from 1 = Not effortful at all to 7 = Very effortful [10]). Demographic information was collected from participants upon concluding the main experiment.

**Result.** *Manipulation check*. An ANOVA on manipulation check of cognitive load indicated participants in high cognitive load perceived higher load (M = 4.94, SD = 1.35) than

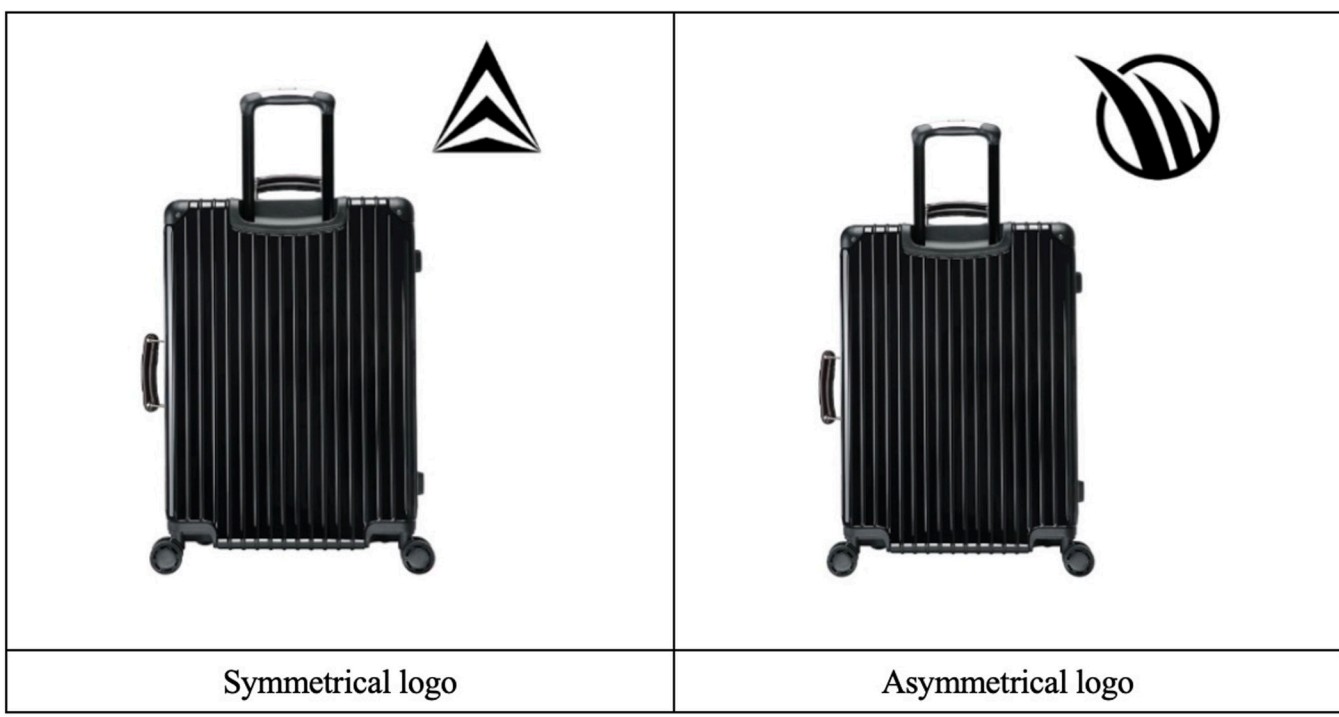

**Fig 5. Stimuli using in Study 3.**

those in low cognitive load (M = 3.35, SD = 1.49; F(1, 197) = 62.459, p < .001), suggesting a successful manipulation.

*Perceived quality*. A 2×2 ANOVA on participants' perceived product quality revealed significant main effect of cognitive load (F (1, 195) = 9.663, p = .002, η² = .047) and their interaction effect (F (1, 195) = 7.383, p = .007, η² = .036). However, the main effect of logo symmetry was not significant (F (1, 195) = 1.428, p = .233). Specifically, when the participants' cognitive load was low, participants in symmetrical logo condition had higher perceived quality (M = 4.83, SD = 1.06) than those in asymmetrical logo condition (M = 4.09, SD = 1.37; F (1, 98) = 8.952, p = .004). However, this effect was dismissed when the participants' cognitive load was high (F (1, 97) = 1.009, p = .318, Fig 6).

**Discussion.** Study 3 examined the boundary condition of the effect of logo symmetry on perceived quality, specifically cognitive load. When consumers are under high cognitive load, the impact of logo symmetry on perceived quality may disappear due to limited cognitive resources. This effect is only observed when consumers are in a low cognitive load condition. In addition, Study 3 employed stimuli different from those in Studies 1 and 2 to further assess the generalizability of logo symmetry's effect on perceived quality across various product categories. The three studies tested the research hypotheses across diverse stimulus conditions. Overall, empirical support was found for Hypotheses 1, 2, and 3.

## General discussion

This research conducted three studies to investigate the effect of logo symmetry on consumers' perceived quality. Study 1 confirmed that symmetrical logos significantly enhance consumers' perception of product quality compared to asymmetrical logos. Study 2 explored the underlying mechanism, revealing that symmetrical logos increase perceived stability, which in turn

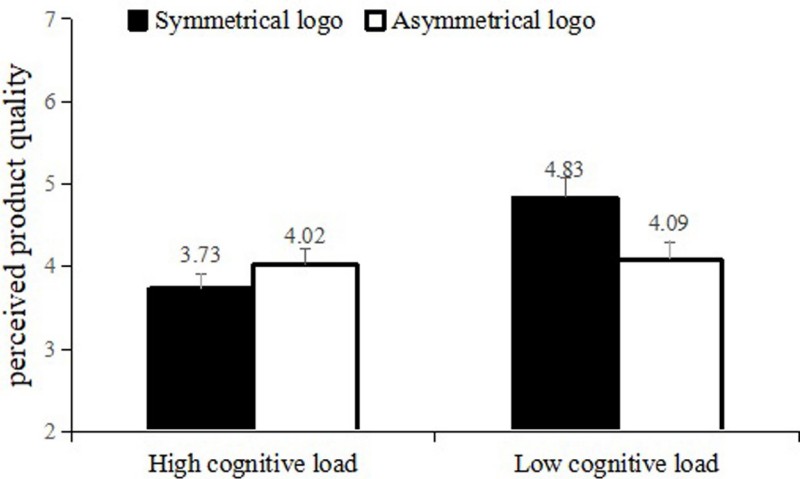

**Fig 6. The result of Study 3.**

enhances perceived product quality. Study 3 examined the moderating effect of cognitive load, showing that the impact of logo symmetry on perceived quality is only present under low cognitive load. When consumers are under high cognitive load, this effect disappears.

## Theoretical contribution

This study offers several theoretical contributions to existing research. First, this study expands the literature on visual perception of logos. Previous studies have considered elements such as the angle and curvature of logos [10], logo color, and logo dynamism, but research on the symmetry of logo design has been relatively sparse. Existing studies primarily focus on the aesthetic and symbolic significance of symmetrical logos [9], their positive impact on consumer product expectations [15], and the specific values they convey [4]. Few studies directly examine the impact of logo design on product attributes. This study extends this line of inquiry by demonstrating that logo design can directly influence consumers' perceptions of product attributes, providing new directions for logo-related attributes and product design research.

Second, this study introduces perceived stability as a mediating variable, broadening the theoretical application of context. Previous research on perceived stability has primarily focused on price or numerical effects [38], such as how unstable prices or the perceived stability of round numbers [26] influence consumer behavior. This study brings perceived stability into the realm of logo design, showing that logo design can impact consumers' perceived stability, which significantly influences their perception of product attributes.

Third, this study provides a theoretical framework and delineates its boundary conditions. The impact of logo design on consumers' perception of product attributes may be influenced by cognitive processes. When consumers are under high cognitive load, the effect of logo design on their perception of product attributes may diminish or even disappear. This finding aligns with Jiang et al. (2016), who showed that external cognitive load affects consumers' cognitive processes [10], further enhancing the theoretical framework within the field of logo design research.

Finally, this study also emphasizes both similarities and differences with previous literature. While prior studies highlighted the aesthetic and symbolic role of symmetry [9, 14], this research extends these findings by showing a direct impact on perceived product quality

through perceived stability. Additionally, while the concept of perceived stability has been discussed in other contexts such as price [38] and round numbers [26], this work uniquely applies it to visual design, thereby broadening the scope of perceived stability in consumer behavior research.

## Practical contribution

This study offers several practical implications for businesses. First, this study suggest that companies can enhance consumers' perceived quality of their products by designing symmetrical logos. This strategy is particularly effective for prevention-oriented products, which need to convey a strong sense of stability to consumers. Designing a symmetrical logo can significantly increase consumers' trust in these products, thereby boosting their purchase intentions. Previous research indicates that symmetrical logos enhance perceptions of product stability through a visual sense of balance, directly relating to consumers' feelings of safety and trust [39]. Therefore, companies can use symmetrical designs to strengthen their product's market competitiveness and consumer purchase intentions.

Second, this study provides clear guidance on highlighting logo features when promoting products. Companies can emphasize the symmetry of their logos in marketing efforts to create positive perceptions of product attributes in consumers' minds [9]. This strategy can enhance market appeal and significantly increase purchase intentions. By prominently displaying the symmetrical logo in advertisements, packaging, or promotional materials, companies can communicate product stability, thereby enhancing consumer trust and purchase intentions. Additionally, this emphasis plays a crucial role in brand building. Consistently highlighting logo symmetry helps establish a unified brand image associated with high quality and stability, improving both individual product performance and overall brand competitiveness.

Third, this study considered an effective boundary condition for businesses: cognitive load. When consumers face complex external environments, such as using social media [40], they experience increased cognitive load. Prior cognitive load negatively impacts the influence of readily available information on decision-making, such as brand choice and product similarity ratings [41]. Companies should avoid media that impose high cognitive load when promoting products, as consumers may not pay attention to the information conveyed by the logo under such conditions.

Finally, based on the findings of this study, policymakers should support initiatives aimed at promoting effective visual branding, especially for businesses involved in consumer goods. Encouraging companies to adopt symmetrical logo designs can enhance consumer perceptions of product quality and stability, thereby contributing to overall market efficiency. Additionally, regulatory bodies could provide guidelines on marketing practices that minimize cognitive load for consumers, ensuring that important product information is effectively communicated [42]. This could involve setting standards for advertising in environments with high cognitive complexity, such as social media platforms.

## Limitation and future research

Although this study provides several theoretical and practical contributions, there are certain limitations. First, the experiments were primarily conducted online, not considering the more complex influences of real-world environments on consumer perceptions. Logos do not exist independently in reality, and future research could explore how different environments affect the perception of product attributes influenced by logo design. Additionally, this study used black-and-white logos to directly test the impact of logo symmetry, but real-world logos often incorporate various colors. Previous research has shown that logo color can also affect

consumer perceptions [43]. Future research could further investigate the interaction between logo color and logo symmetry.

Another limitation is the use of a controlled setting that may not fully capture the nuances of consumer decision-making in natural environments. Future research could employ field studies or real-world experiments to examine how logo symmetry impacts perceived quality in different consumer settings, such as retail stores or digital marketplaces. Moreover, future research could explore other visual elements, such as complexity [44], curvature [45], and dynamism [46], and their combined effects on consumer perception. The cross-cultural applicability of these findings is another area for further investigation, as visual preferences and cognitive processing may vary significantly across different cultural contexts.

## Supporting information

**S1 File. Overview of study.**
(DOCX)

**S2 File. Data of Study 1.**
(XLSX)

**S3 File. Pretest of Study 1.**
(XLSX)

**S4 File. Data of Study 2.**
(XLSX)

**S5 File. Data of Study 3.**
(XLSX)

**S6 File. Pretest of Study 3.**
(XLSX)

**S7 File.**
(XLSX)

## Author Contributions

**Conceptualization:** Xianghu Wu.

**Writing – original draft:** Xianghu Wu.

**Writing – review & editing:** Xianghu Wu.

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
