## [Decision Letter · Decision Letter 0]

5 Nov 2024

PONE-D-24-32602Beauty of Symmetry - The Impact of Logo Symmetry on Perceived Product QualityPLOS ONE

Dear Dr. Wu,

Thank you for submitting your manuscript to PLOS ONE. After careful consideration, we feel that it has merit but does not fully meet PLOS ONE’s publication criteria as it currently stands. Therefore, we invite you to submit a revised version of the manuscript that addresses the points raised during the review process.

We look forward to receiving your revised manuscript.

Kind regards,

Seda Yildirim, PhD

Academic Editor

PLOS ONE

3. Please ensure that you include a title page within your main document. You should list all authors and all affiliations as per our author instructions and clearly indicate the corresponding author.

Additional Editor Comments:

This study can potentially contribute to the literature. However, it should be made more scientific with some corrections.

First of all, the introduction seems short. The introduction should be expanded and the main contribution of the study should be stated.

The first person singular we and i should not be used in the study. There should be expressions such as this study or in this study instead of we and our.

The conclusion of the study should be written in more detail with previous literature. The differences or similarities between the previous literature and the current literature are emphasized. More policy recommendations should be given. limitations and future directions should be stated.

Also, all text must be rearranged for stitching.

Hypothesis setups should be revised. Hypotheses should be established with more scientific and clear sentences.

Reviewers' comments:

Reviewer's Responses to Questions

**Comments to the Author**

1. Is the manuscript technically sound, and do the data support the conclusions?

Reviewer #1: Yes

Reviewer #2: Yes

2. Has the statistical analysis been performed appropriately and rigorously? 

Reviewer #1: Yes

Reviewer #2: Yes

3. Have the authors made all data underlying the findings in their manuscript fully available?

Reviewer #1: No

Reviewer #2: Yes

4. Is the manuscript presented in an intelligible fashion and written in standard English?

Reviewer #1: No

Reviewer #2: Yes

5. Review Comments to the Author

Reviewer #1: This article investigates the effect of logo symmetry on perceived product quality, and as a scientific reviewer, it can be evaluated from the following perspectives:

Literature Review and Theoretical Foundation: The article provides a detailed literature review supporting the effect of logo symmetry on perceived product quality. However, it would benefit from referencing more recent studies and discussing the influence of other logo design elements in greater depth beyond symmetry.

Hypothesis Development: The hypotheses are clearly stated, yet the hypothesis that "symmetrical logos increase perceived quality" could be grounded in a more concrete theoretical framework. Strengthening theoretical explanations, particularly regarding symmetry’s impact on cognitive processing alongside its aesthetic value, would be beneficial.

Methodology and Experimental Design: The three experiments used in the study offer an appropriate methodological approach to test the hypotheses. However, expanding the participant pool to increase the generalizability of the findings would be recommended. Additionally, the exclusive use of online experiments may overlook the complex influences present in real-world environments.

Presentation of Findings: The findings are clearly presented, but further interpretation of the statistical analyses could be useful. In particular, detailing the role of cognitive load would help readers better understand this factor’s influence.

Theoretical and Practical Contributions: The article articulates its theoretical contributions well, though it would be beneficial to support the practical recommendations with specific application examples. This would provide a more concrete guide for businesses regarding logo design considerations.

Conclusion and Limitations: While the article summarizes its research findings and offers suggestions for future studies, emphasizing the limitations that might be encountered in real-world applications could further enhance the reliability of the study.

In conclusion, the article has a generally strong structure; however, it could be enriched by adding more depth in areas such as the literature review, theoretical framework, methodological details, and practical contributions, making it scientifically more robust.

Reviewer #2: A few explanatory sentences about the symmetry effect can be added to the introduction.

The discussion sections of the study should be expanded in a more explanatory manner.

The general discussion section should also be expanded.

6. PLOS authors have the option to publish the peer review history of their article (what does this mean?). If published, this will include your full peer review and any attached files.

Reviewer #1: No

Reviewer #2: No

---

## [Author Response · Author response to Decision Letter 0]

2 Dec 2024

Responses to the Editor and Reviewers’ Comments

Manuscript ID PONE-D-24-32602

Dear Editor and Reviewers:

Thank you for your letter and for the reviewers’ comments concerning our manuscript entitled “Beauty of Symmetry - The Impact of Logo Symmetry on Perceived Product Quality” (ID: PONE-D-24-32602). These comments were very helpful for revising and improving our paper as well as for guiding our research. We have studied the comments carefully and have made corrections that we hope are satisfactory. The revised text is marked with different colour in the paper. We have indicated the location of the revisions, including the line and page numbers, to make finding the - Changed/added content easier in the manuscript. Overall, We have enhanced the introduction and conclusion sections and added more evidence to support our research. Furthermore, we have revised the formatting of the entire paper to eliminate grammatical errors.

According to the editor's guidance, we have replied to each reviewer's comments in a 3-column table.

Responses to Additional Editor

# Additional Editor Comments

This study can potentially contribute to the literature. However, it should be made more scientific with some corrections.

- Response to the comment:

I appreciate your positive and constructive comments regarding our manuscript. Your detailed comments have provided us with clear directions for modification.

Below, I respond to each of your comments. We have made every effort to revise and improve the manuscript according to your comments. I hope that our - Responses and the revised manuscript meet your expectations.

- Changed/Added in the manuscript:

N/A

#comment 1:

First of all, the introduction seems short. The introduction should be expanded and the main contribution of the study should be stated.

- Response to the comment:

I appreciate your positive and constructive comments regarding our manuscript. Your detailed comments have provided us with clear directions for modification.

- Changed/Added in the manuscript:

I have added contents to the “Introduction”. The details are as follows :

“One of the key factors in logo design is symmetry. Symmetrical designs are often perceived as more attractive, balanced, and credible (Cárdenas and Harris, 2006). This preference for symmetry is rooted in cognitive processing; symmetrical visuals are easier to process, leading to a perception of harmony and stability (Bertamini and Makin, 2014). Therefore, exploring how logo symmetry influences consumers' perception of product quality is of both theoretical and practical importance, as it can help businesses create logos that foster positive consumer evaluations and brand loyalty (Bettels and Wiedmann, 2019).”

“This study seeks to address this gap by examining how logo symmetry influences consumers' perception of product quality.”

“Our findings contribute to the literature on logo design and visual perception in several ways. First, this study demonstrate that logo symmetry directly impacts consumer perception of product attributes, providing insights into how visual elements influence consumer evaluations beyond mere aesthetics. Second, this study introduce perceived stability as a mediating factor, expanding the understanding of how logo design affects consumer behavior. Finally, this study highlight the role of cognitive load in moderating the effect of logo symmetry, suggesting that the effectiveness of visual design elements can vary depending on the cognitive context of consumers.

From a practical perspective, this study provides actionable insights for marketers and designers. Companies can enhance consumers' perceived quality of their products by adopting symmetrical logos, especially for products that need to convey stability and trust (Bettels and Wiedmann, 2019). Furthermore, marketing strategies should consider the cognitive load of the target audience - symmetrical logos are more effective when consumers are not overwhelmed by competing stimuli. This study also opens avenues for future studies to explore the impact of other visual elements, such as color and shape complexity, on consumer behavior, particularly across different cultural contexts. In summary, by investigating the impact of logo symmetry on perceived product quality and the mechanisms behind this effect, this study aims to advance the understanding of logo design's role in consumer perception. These findings are not only theoretically significant but also offer practical guidance for effective marketing and branding strategies in today’s visually driven marketplace.”

(Section “Introduction” on Line 27 - 33, Line 44 - 45, Line 51 - 69)

#comment 2:

The first person singular we and i should not be used in the study. There should be expressions such as this study or in this study instead of we and our.

- Response to the comment:

Thank you for your corrections regarding the wording in this study. I now realize that first-person singular should be avoided in describing this research, and I have revised all expressions that were inconsistent with this convention.

- Changed/Added in the manuscript:

I have changed “I”, “We”, and “Our” into “This Study”

#comment 3:

The conclusion of the study should be written in more detail with previous literature. The differences or similarities between the previous literature and the current literature are emphasized. More policy recommendations should be given. limitations and future directions should be stated.

- Response to the comment:

Thank you for your constructive suggestions regarding the study’s limitations in the conclusion. These insights are invaluable for strengthening the manuscript. Based on the recommendations, I have expanded the conclusion section to include comparisons with existing literature and additional policy suggestions. Furthermore, we have considered other potential research directions, thereby increasing this study’s relevance as a reference for future research.

- Changed/Added in the manuscript:

I have added contents to the manuscript. The details are as follows :

“Finally, this study also emphasizes both similarities and differences with previous literature. While prior studies (e.g., Bettels and Wiedmann, 2019; Krey and Rossi, 2018) highlighted the aesthetic and symbolic role of symmetry, this research extends these findings by showing a direct impact on perceived product quality through perceived stability. Additionally, while the concept of perceived stability has been discussed in other contexts such as price (Turnovsky et al., 1980) and round numbers (Pena‐Marin and Bhargave, 2016), this work uniquely applies it to visual design, thereby broadening the scope of perceived stability in consumer behavior research.”

(Section “Theoretic Contribution” on Line 429 - 435)

“Finally, based on the findings of this study, policymakers should support initiatives aimed at promoting effective visual branding, especially for businesses involved in consumer goods. Encouraging companies to adopt symmetrical logo designs can enhance consumer perceptions of product quality and stability, thereby contributing to overall market efficiency. Additionally, regulatory bodies could provide guidelines on marketing practices that minimize cognitive load for consumers, ensuring that important product information is effectively communicated (Drichoutis, 2017). This could involve setting standards for advertising in environments with high cognitive complexity, such as social media platforms.”

(Section “Practical Contribution” on Line 465 - 472)

“Another limitation is the use of a controlled setting that may not fully capture the nuances of consumer decision-making in natural environments. Future research could employ field studies or real-world experiments to examine how logo symmetry impacts perceived quality in different consumer settings, such as retail stores or digital marketplaces. Moreover, future research could explore other visual elements, such as complexity (Mahmood et al., 2019), curvature (Palumbo and Bertamini, 2016), and dynamism (Cian et al., 2014), and their combined effects on consumer perception. The cross-cultural applicability of these findings is another area for further investigation, as visual preferences and cognitive processing may vary significantly across different cultural contexts.”

(Section “Limitation and Future Research” on Line 484 - 492)

#comment 4:

Also, all text must be rearranged for stitching.

- Response to the comment:

Thank you for your valuable suggestions on this manuscript. I have revised it to align with the formatting and layout guidelines required by PLOS ONE, ensuring full compliance with their standards. Additionally, I have expanded the research content in response to your earlier recommendations.

- Changed/Added in the manuscript:

Please refer to the manuscript for detailed revisions, which have been highlighted in different color for easy identification.

#comment 5:

Hypothesis setups should be revised. Hypotheses should be established with more scientific and clear sentences.

- Response to the comment:

Thank you for your suggestions on the hypothesis setups section. Based on your feedback, I have refined the hypothesis development section by adopting more rigorous academic language and adding real-world examples to improve the clarity and accessibility of this study.

- Changed/Added in the manuscript:

I have changed contents to the section Hypothesis Development. The details are as follows :

“Symmetrical logos are fundamental in enhancing visual stability, making them inherently more appealing and leading to a more positive perception among consumers. Empirical studies have consistently demonstrated that symmetrical designs are perceived as more aesthetically pleasing, thereby significantly influencing consumer interactions with logos and brand elements (Bajaj & Bond, 2017). The concept of symmetry extends beyond mere visual attractiveness; it conveys attributes such as balance, order, and stability, which collectively contribute to a positive perception of a brand's intrinsic values. This preference for symmetry indicates that symmetrical logos evoke a sense of coherence, elegance, and high quality, which ultimately leads to more favorable consumer attitudes and a deeper emotional connection with the brand (Creusen et al., 2010). For example, the logos of brand like Toyota is quintessential example of symmetry that convey these positive associations, contributing to their strong brand identities. Furthermore, the consistent use of symmetrical logos fosters brand identity, reinforcing consumer trust and loyalty by evoking a sense of familiarity and stability (Coelho et al., 2019). In addition, symmetry is frequently associated with permanence and reliability—key attributes that are crucial for brands seeking to establish long-term relationships with consumers, such as in the case of the symmetrical logo of McDonald's, which conveys reliability and endurance (Pena‐Marin & Bhargave, 2015).

A substantial body of literature has demonstrated that stable visual elements significantly enhance consumer perceptions of product quality, durability, and shelf life. For instance, Coelho et al. (2019) illustrate how stable shapes contribute not only to a positive brand image but also to an enhanced perception of product quality, while Pena‐Marin and Bhargave (2019) argue that products perceived as having stable attributes are evaluated more favorably by consumers. This can be observed in the packaging of high-end perfumes, which often use sturdy, symmetrical bottles to convey a sense of premium quality and stability. Similarly, Yan et al. (2013) noted that smaller, meticulously designed products are often seen as more challenging to manufacture, thereby implying higher quality. Applying this concept to stable shapes, products that visually signal stability—such as the iconic Coca-Cola glass bottle—are often perceived as having higher quality. Mitra & Golder (2006) highlight the impact of objective quality on consumer perceptions, emphasizing that stability in design fosters consumer trust and loyalty (Leischnig & Enke, 2011). For example, the durable and stable design of Tupperware products conveys longevity and reliability, which positively influences consumer perceptions of quality and durability. In conclusion, products with stable visual elements are more likely to enhance consumer perceptions of quality, significantly shaping evaluations of product longevity and fostering a positive brand image. ”

(Section “Hypothesis Development - Logo Symmetry and Perceived Quality” on Line 130 - 163)

“Cognitive load refers to the mental effort required to process information in working memory. When cognitive load is low, individuals have greater cognitive resources available to engage with and appreciate complex stimuli, such as the symmetry of logos (Sweller, 1988). Conversely, under conditions of high cognitive load, these cognitive resources are largely occupied, which can hinder the ability to process intricate visual details. For example, consumers shopping online during a relaxed browsing session (low cognitive load) may be more capable of appreciating a brand logo's symmetrical design than when multitasking or in a rushed environment (high cognitive load). Research suggests that high logo symmetry is associated with higher perceived quality under low cognitive load, whereas this effect diminishes under high cognitive load. This is supported by findings that cognitive load influences visual perception and attention mechanisms. Lavie et al. (2004) and Konstantinou and Lavie (2013) demonstrated that individuals have greater capacity to process visual details, such as symmetry, under low cognitive load conditions, leading to enhanced perception and evaluation of these stimuli. Moreover, empirical studies by Saiphoo and Want (2018) and Derpsch et al. (2021) indicate that cognitive load plays a significant role in shaping the evaluation of visual information, further supporting the notion that symmetry is more effectively appreciated when cognitive resources are not heavily taxed. In a practical context, this suggests that brands aiming to convey a sense of quality through logo design should consider the cognitive state of their target audience. For example, advertising campaigns during times when consumers are likely to be relaxed (e.g., during weekends or leisure activities) may enhance the positive effect of symmetrical logos on perceived quality.”

(Section “Hypothesis Development - Moderator role of cognitive load” on Line 171 - 190)

Responses to Reviewer #1

#overall:

This article investigates the effect of logo symmetry on perceived product quality, and as a scientific reviewer, it can be evaluated from the following perspectives

- Response to the comment:

Thank you for your constructive feedback on this study. I have revised the entire manuscript according to your suggestions.

- Changed/Added in the manuscript:

N/A

#comment 1:

Literature Review and Theoretical Foundation: The article provides a detailed literature review supporting the effect of logo symmetry on perceived product quality. However, it would benefit from referencing more recent studies and discussing the influence of other logo design elements in greater depth beyond symmetry.

- Response to the comment:

I also appreciate your acknowledgment of the literature review and your guidance on addressing certain limitations. In the “Limitations and Future Research” section, I have added a summary of current findings on logo symmetry and discussed related research gaps.

- Changed/Added in the manuscript:

I have added contents to the manuscript. The details are as follows :

“Moreover, future research could explore other visual elements, such as complexity (Mahmood et al., 2019), curvature (Palumbo and Bertamini, 2016), and dynamism (Cian et al., 2014), and their combined effects on consumer perception. The cross-cultural applicability of these findings is another area for further investigation, as visual preferences and cognitive processing may vary significantly across different cultural contexts.”

(Section “Limitation and Future Research” on Line 484 - 492)

#comment 2:

Hypothesis Development: The hypotheses 

---

## [Editor Report · Decision Letter 1]

26 Dec 2024

Beauty of Symmetry - The Impact of Logo Symmetry on Perceived Product Quality

PONE-D-24-32602R1

Dear Dr. Wu,

We’re pleased to inform you that your manuscript has been judged scientifically suitable for publication and will be formally accepted for publication once it meets all outstanding technical requirements.

Kind regards,

Seda Yildirim, PhD

Academic Editor

PLOS ONE
---

## [Editor Report · Acceptance letter]

16 Jan 2025

PONE-D-24-32602R1 

PLOS ONE

Dear Dr. Wu, 

I'm pleased to inform you that your manuscript has been deemed suitable for publication in PLOS ONE. Congratulations! Your manuscript is now being handed over to our production team.

Kind regards, 

on behalf of

Professor Seda Yildirim 

Academic Editor

PLOS ONE